# Current Perspectives on the Use of off the Shelf CAR-T/NK Cells for the Treatment of Cancer

**DOI:** 10.3390/cancers13081926

**Published:** 2021-04-16

**Authors:** Lauren C. Cutmore, John F. Marshall

**Affiliations:** Barts Cancer Institute, Cancer Research UK Centre of Excellence, Queen Mary University of London, London EC1M 6BQ, UK; l.c.cutmore@qmul.ac.uk

**Keywords:** CAR T cell, immunotherapy, adoptive cell transfer

## Abstract

**Simple Summary:**

CAR T cells are a type of immunotherapy whereby a patient’s own cells are genetically modified to recognise and kill the patient’s own cancer cells. Currently, each patient has CAR T cells made from their own blood cells. This type of therapy has had a big impact on the treatment of blood cancers, however making an individual treatment from each patient is expensive and labour intensive. This review discusses the potential of making CAR T cells more widely available by producing them in large numbers from healthy donors.

**Abstract:**

CAR T cells have revolutionised the treatment of haematological malignancies. Despite this, several obstacles still prohibit their widespread use and efficacy. One of these barriers is the use of autologous T cells as the carrier of the CAR. The individual production of CAR T cells results in large variation in the product, greater wait times for treatment and higher costs. To overcome this several novel approaches have emerged that utilise allogeneic cells, so called “off the shelf” CAR T cells. In this Review, we describe the different approaches that have been used to produce allogeneic CAR T to date, as well as their current pre-clinical and clinical progress.

## 1. Introduction

It has long been recognised that the immune system plays a vital role in the establishment and progression of tumours [1,2,3]. In healthy individuals, the immune system seeks out and destroys malignant cells before they form established tumours. Nevertheless, tumours do form in immunocompetent individuals, and the possible process by which this develops is described by the immunoediting hypothesis [4]. Briefly, it comprises three phases: elimination, equilibrium, and escape. During the Elimination phase the immune system destroys transformed cells faster than they can grow back. The Equilibrium phase sees the rate of destruction of transformed cells matching their proliferative capacity. Finally, the selective pressure of immune attack results in selection for and growth of resistant cells who exhibit immune escape and the tumour progresses. Due to the pivotal role of the immune system in cancer progression, immunotherapy has again attracted significant attention in the last few years. Immunotherapy modulates the patients’ immune system to promote or enhance cancer cell recognition and killing. Immunotherapy can be split into two areas, active immunotherapies, which further stimulate the patients’ existing effector function, and passive therapies, which compensate for a lack of host effector function.

Active immunotherapies, such as checkpoint inhibitors, have failed to bring about the population-wide improvement in cancer therapy that was hoped for but have generated impressive responses in a much smaller fractions of treated patients [5,6]. This is probably due to a combination of factors that are becoming clearer as these disappointing responses have arisen. 

Mutational load present in the tumour is predictive of immune checkpoint blockade response, with patients with more mutations or microsatellite-instability having improved survival and more durable responses [7]. Similarly, mismatch repair deficiency can result in greater responses to checkpoint inhibitors, possibly due to the enhanced number of neo-antigens produced as a result of such deficiency [8]. More recently, the gut microbiome has been implicated in the response to active immunotherapies. The microbiome plays a role in activating the innate immune system against the tumour [9,10]. MHC I downregulation is prevalent in many cancers and allows the cancer cells to evade immune cell recognition, even in combination with checkpoint inhibitors [6]. The HLA subtype expressed by a patient can also affect the efficacy of active immunotherapies, especially cancer vaccines [11]. Tumour infiltrating lymphocyte (TIL) infiltration present in the tumour is also predictive of positive response to active immunotherapies [12,13,14,15,16].

To overcome this lack of active effector cells within the patient, adoptive cell therapies (ACT) have been investigated, whereby effector cells are introduced to the patient to destroy the cancer [17]. The earliest studies from Steve Rosenberg investigated the systemic administration of lymphokine-activated killer cells (patient lymphocytes cultured ex-vivo in the presence of IL-2 then re-infused) and IL-2 in patients with metastatic cancer. This resulted in some partial responses and one patient with complete tumour regression [18].

In later studies, tumour infiltrating lymphocytes (TIL) were isolated from patients and expanded ex-vivo before reinfusing them back into the same patient at higher numbers [19]. The first clinical trial using TILs to treat metastatic melanoma showed an overall objective response rate of 34%. Side effects included oedema, respiratory distress and disorientation, but are thought to have been partially caused by the IL-2 administered as part of the treatment regimen [20]. To improve the potency of the T cell response and to redirect them towards a specific target, TCR therapy and chimeric antigen receptor (CAR) T cells were developed [21]. CAR T cells are T cells that are modified to express an artificial T cell receptor called a chimeric antigen receptor (CAR) that is designed to redirect the T cell towards a specific cancer cell antigen.

Conventionally CAR T cells are produced from the patient’s own T cells, isolated from the peripheral blood, genetically modified to express a CAR with specificity for a designated target antigen and then reinfused into the patient following lymphodepletion. These cells can recognise cell surface antigens in an MHC- independent manner allowing a wider range of antigens to be targeted. The structure of the CAR differs to the conventional T cell receptor (TCR) that is composed of a TCR heterodimer. CARs are composed of several distinct regions each with a unique function [22]. The extracellular domain of the CAR contains antigen recognition domain that is able to bind to the target expressed on the cancer cells. This binding domain is typically a single chain variable fragment (ScFv) against the target but can also be derived from a ligand or peptide (such as the A20FMDV2 peptide that binds to αvβ6) that binds specifically to the antigen [23,24]. This domain is followed by a hinge region that provides physical flexibility for the CAR and facilitates the formation of the immune synapse [25]. The transmembrane domain connects the extracellular domain of the receptor with the intracellular domain and allows signal transduction into the cell. The intracellular domain composition represents the most varied part of the CAR as it is mostly here that the various generations of CARs vary between each other. Thus the first generation CARs only contained a single signalling domain, usually CD3 zeta, which is responsible for the cytotoxic activity of the cell [26]. Subsequent generations of CARs also contain one or more co-stimulatory domains (usually 4–1BB or CD28) which provided pro-survival signals and improved persistence of the CAR T cells [27].

## 2. Autologous and Allogeneic CAR T Cells

CAR T cells have revolutionised the treatment of haematological malignancies, with three CD19 targeting CARs licenced for use [28,29,30]. However, these successes are yet to be translated into solid cancers. One of the obstacles to scalable CAR T cell treatment of both liquid and solid cancers is the requirement of autologous T cells, which imposes several problems. Firstly due to the small scale production, the cost per patient is extremely high, up to $475,000 [31]. This also makes the production extremely labour intensive as each patient must have CAR T cells made individually for them. Due to this personalised approach the manufacturing time for CAR T cell is around three weeks [32]. This is often too long for patients to wait for the treatment and by the time they receive the CAR T cells, significant disease progression may have occurred [33]. Moreover, patients with advanced cancer are often lymphopenic due to disease burden and prior therapy, with residual T cells of poor quality and function [34]. This effect is compounded by the fact that T cells from these patients may have also been in contact with the immunosuppressive tumour microenvironment [34]. In the ELIANA study of a CD19 targeting CAR for B-ALL, 7.6% of patients did not receive the CAR T cells due to ‘product related issues’, whereby it was not possible to manufacture a sufficient number of CAR T cells for infusion [35]. In another trial of a CD19 CAR for the treatment of B-cell Precursor Acute Lymphoblastic Leukemia 9% of patients did not receive CAR T cells product due to manufacturing issues [36]. This illustrates the issues with using autologous T cells for effective anti-cancer therapy.

The use of different cells for each recipient, namely autologous cells from each patient, can result in a large variation between the quality of the CAR T cells produced, making dosing and efficacy hard to compare. Patients who don’t respond to CAR T cell therapy tend to have received products with a more differentiated effector T cell phenotype, whereas those who respond better often have an early memory phenotype [37]. In addition, expression of certain receptors on the T cell used to produce the CAR T can confer a more efficacious product; for example, expression of the IL-6 receptor is favourable [37]. The low number of CAR T cells made from autologous donation also means that re-dosing with the same product is not usually possible and patients must undergo the same process multiple times if repeated administration is required.

Conceptually, these obstacles can be overcome by using allogeneic T cells as the basis for producing CAR T cells. This “off the shelf product” made from donor T cells could be produced on an industrial scale from donors with an ideal T cell phenotype. This would produce a superior product that could be administered immediately with easy re-dosing. This would also allow patients to be treated with multiple CAR T cells simultaneously, targeting different antigens, to improve efficacy and prevent tumour resistance. Above all it would greatly reduce the manufacturing costs making it more accessible to patients [38].

However, there are also some difficulties inherent in the development of allogeneic CAR T cell products. The main hurdle is graft versus host disease (GvHD) which occurs when the allogeneic T cells recognise the host’s cells as foreign and mount an immune response against them. This is mediated through the action of the allogeneic T cell receptor against the host HLA. GvHD can be fatal and has been a major obstacle to successful allogeneic haematopoietic stem cell transplantation (HSCT) [39].

Conversely, the host immune system may recognise the donor T cells as foreign causing rejection and destruction of the CAR T cells, terminating the therapeutic effect. One of the main requirements for effective CAR T cell immunotherapy is persistence of the infused cells within the patient. CD19 CAR T cells have been detected in the blood of patients several years after treatment [39,40]. Due to the foreign nature of allogeneic T cells, they have poorer persistence in vivo and consequently may not result in sustained responses. To allow allogeneic T cells to survive and have anti-cancer effects in the patient, several approaches can be taken. A list of ongoing allogeneic CAR T cell clinical trials is provided in Table 1. Approaches to develop off the shelf CAR cell therapies are discussed in greater detail in the sections that follow.

## 3. γδ. T Cells

An emerging field is the use of different T cell subtypes to produce CAR T cells. Conventionally αβ T cells have been used for production of CAR T cells, however, more recently there has been significant interest around using γδ T cells instead as they have advantages over αβ T cells.

γδ T cells represent a small percentage of total T cells, around 2% of peripheral blood T cells [41,42], however they constitute a large proportion of tissue resident T cells, especially in the gut mucosa [43]. The TCR heterodimer of γδ T cells is made up of a γ subunit and a δ subunit in contrast to the α and β subunit present in αβ T cells. Both T cell types are capable of strong cytotoxic effector function via both death receptor interactions and release of cytolytic granules [44]. Additionally, they can release a wide range of inflammatory cytokines including TNFα and IFNγ [44]. Notably γδ T cells are also able to process and present antigens to CD4 and CD8 T cells in a similar manner to professional antigen presenting cells (APC) [45]. There are two main subtypes of γδ T cell. Vδ2 T cells are the most common subtype and constitute the majority of circulating γδ T cells. In contrast, non-Vδ2 cells are more common in mucosal sites [46,47]. Subsets of both αβ and γδ T cells are able to recognise lipid antigens presented on CD1 proteins via their respective TCR [48,49,50].

γδ T cells are a key component of the innate immune system that help to detect and eradicate tumour cells [51]. The γδ TCR recognise unprocessed antigens and do not rely on MHC presentation for antigen recognition in contrast to most αβ T cells, which recognise processed peptide antigens presented on MHC-I [52]. This means γδ T cells can target many different antigens not recognised by the αβ subset. Additionally, they show less toxicity towards healthy cells than αβ T cells [53,54].

Similarly to NK and CD8+ αβ T cells cells, γδ T cells express NKG2D on their surface. This receptor is key in identifying malignant cells and provides a co-stimulatory signal via DAP10 [55]. NKG2D recognises several UL16-binding proteins, which are upregulated in stressed and transformed cells [56], as well as an MHC class I polypeptide-related sequence A and B (MICA/B) (Figure 1) [57]. This allows the cells to identify transformed cells in an alternative way to expression of a specific antigen. DNAM-1(DNAX accessory molecule-1) is another activation receptor expressed on γδ T cells that bind to poliovirus receptor (PVR/CD155), and Nectin-2 ligands, which are commonly upregulated on tumour cells (Figure 2) [58,59]. The Vγ9Vδ2 T cell receptor is expressed on the main circulating γδ T cell subset and is capable of identifying phosphorylated antigens (pAgs), products of mevalonate metabolism that are frequently overproduced in transformed and infected cells (Figure 2) [60]. Not only cell surface antigens are targetable with γδ T cells, soluble proteins within the TME can also be recognised, further increasing their range of action (Figure 1) [61]. Further arming the γδ T cells with a CAR construct can allow targeting of specific cell surface antigens and increased cytotoxicity [62]. Activated γδ T cells induce a wide range of immune responses, recruiting αβ T cells, APCs and B cells and providing costimulatory signals to NK cells [63,64].

Despite the small starting number of γδ T cells present in peripheral blood, these cells can be expanded ex vivo to produce clinically significant numbers that can be used therapeutically [65]. They are unlikely to induce GvHD because the activation of their TCR is not MHC restricted. CAR γδ T cells directed towards the disialoganglioside GD2, which is frequently overexpressed in gliomas and other tumours of neuroectodermal origin, enhance cytotoxicity against GD2-expressing cell lines. γδT expressing a CAR targeting CD19 for leukaemia treatment showed cytotoxicity towards CD19+ tumour cell lines in vitro and in vivo. They also demonstrated a CAR independent effect against CD19- clones [62]. There is currently a study ongoing which is examining the feasibility of γδ T cell expansion from peripheral blood of AML patients and if these cells will tolerate modification with a CAR (NCT03885076). A phase 1 trial investigating the safety of γδ T cells with a NKG2DL-targeting CAR in solid tumours is currently underway (NCT04107142). This study is using both haploidentical and allogeneic γδ T cells (Table 1).

## 4. NK Cells

Natural killer (NK) cells also represent an attractive cellular vehicle for the development of off-the-shelf CAR cell therapy. NK cells play an important role in anti-tumour immunity and exert cytolytic activity against cancer cells via the release of granzyme B and perforin as well as the action of the Fas ligand. NK cells do not express an antigen specific cell surface receptor, but instead have more wide-ranging anti-tumour action mediated by several innate receptors [66]. NK cells express inhibitory receptors such as killer immunoglobulin-like receptor (KIR) which recognise classical HLA class I molecules such as HLA-A to prevent NK cell activity against healthy cells (Figure 2) [67]. Binding of inhibitory KIR to self HLA class 1 prevents NK cell effector function [68,69]. Contrastingly, killer cell activating receptors (KARs) can recognise a host of ligands on tumour cells; for example NKG2C (CD94) recognises non- classical HLA class I molecules including HLA-E (Figure 2) [70]. Tumour cells often downregulate the expression of HLA molecules, resulting in recognition and activation by NK cells.

Another important anti-tumour role of NK cells is via antibody-dependent cell-mediated cytotoxicity (ADCC). The ADCC receptor complex is able to bind with the Fc domain of human immunoglobulin via FcyRIIIA (CD16a) and FcγRIIC/CD32c and cause NK cell activation [71,72,73] (Figure 2). This mechanism allows NK cells to detect and kill tumour cells that have been opsonised with antibodies [74]. This is one of the mechanisms underlying the effectiveness of monoclonal antibody therapy of cancer [75,76]. Activatory receptors cause activation of NK cells, due to the presence of immunoreceptor tyrosine-based activation motifs (ITAMs) in the endodomain of the complexes [77]. ITAMs consist of a tandemly repeated four amino acid motif that provide an important initiating component in the activation of immune cells.

NK cells do not induce GvHD due to the lack of TCR expression and allogeneic NK cells have been successfully transferred to patients without toxicity [78]. They also show a much more favourable toxicity profile compared to adoptively infused CAR T cells, with cytokine storm and neurotoxicity rarely seen [79]. This may be due to differences in cytokine release between T cells and NK cells, but this is not yet fully understood. The MHC1 mismatch between the donor cells and the tumour may also cause increased cytotoxicity against the tumour [80].

Although NK cells have innate anti-cancer activity, further arming them with a CAR directs their action towards a specific antigen. In addition to direct CAR mediated cytotoxicity, NK CARs can also perform (ADCC) and can also recognise cancerous cells via the action of their other cancer-directed receptors (see above) making them an appealing cell type for CAR therapy.

Production of CAR NK cells is much more challenging than CAR T cells due to the lower numbers of NK cells present in the blood and their reduced capacity for expansion [81,82]. Additionally NK cells are also less amenable to genetic manipulation than T cells making them more challenging to work with as an adoptive cell transfer therapy [83]. It is however possible to obtain NK cells from frozen umbilical cord blood (UBC) and this approach has been used to create NK cells expressing a CD19 targeting CAR [84]. This approach has been used in a phase I and II clinical trial which demonstrated the feasibility of the transduction and expansion of NK cells as a therapy. Anti-CD19 CAR-NK cells derived from cord blood were used to treat relapsed or refractory CD19-positive cancers, with 7 out of 11 patients showing a complete response [85].

NK-92 cells, an activated human NK line has also been safely used in patients [86]. The human NK-92 line was derived from a 50-year-old male patient with non-Hodgkin’s lymphoma and has shown cytotoxic activity against both haematological and non-haematological cancer cells in-vitro [87]. NK-92 cells don’t express KIRs and are therefore more active as they don’t receive inhibitory signals from the cancer cells [88]. Just as with primary NK cells, NK-92 cells can be further modified to express CARs to target specific tumour antigens. CD20-targeting CAR NK-92 cells showed efficacy in vitro against lymphoma and leukaemia cells [89]. In a murine model of lymphoma, tumour growth was significantly reduced following treatment with CD20 targeting CAR NK-92 cells compared to those treated with parental NK-92 cells. This also resulted in improved survival in the CAR NK-92 group.

NK-92 cells have also been redirected to target ErbB2 in breast cancer cell lines [90,91]; SKBR3 breast cancer cells were specifically lysed by NK-92- ErbB2 targeting CAR cells compared to NK-92 parental cells in-vitro. NK-92 cells modified to express an anti-PSMA CAR have shown promising results in a preclinical model of prostate cancer. NK-92 cells expressing an anti-PSMA CAR significantly reduced the tumour size and significantly increased survival compared to unmodified NK-92 cells in an orthotropic PC-3 model of prostate cancer [92]. NIH 3T3 fibroblasts expressing oncogenic human ErbB2 treated with NK-92- ErbB2 targeting CAR cells resulted in a reduction in tumour growth in a xenograft model compared to tumours treated with unmodified NK-92 [90].

NK CARs targeting EGFR, derived from both NK-92 cells and primary NK cells from healthy donors, demonstrated activity against glioblastoma cell lines [91]. EGFR-CAR-transduced NK-92 cells have significantly greater cytotoxic activity against EGFR positive glioblastoma cell lines compared to mock-transduced NK cells. These EGFR-CAR NK-92 cells also significantly reduced tumour growth and extended survival in two orthotopic xenograft models of glioblastoma. Primary NK cells transduced with EGFR-CAR also showed enhanced cytotoxicity against patient-derived glioblastoma cell lines in-vitro. CD33 CAR expressing NK-92 cells showed no significant adverse side effects in a phase 1 trial for the treatment of acute myeloid leukaemia (AML) (NCT02944162) [93] (Table 1). Taking the concept of an off-the-shelf CAR even further, a universal CAR NK-92 cell has been developed [94]. This universal CAR (UniCAR) does not target tumour antigens directly, but are redirected using a tumour-specific target module [95]. This method allows the targeting of multiple tumour antigen using the same CAR, by utilizing different targeting modules. When NK-92 expressing the UniCAR were combined with a GD-2 targeting module they specifically lysed neuroblastoma and melanoma cells expressing GD-2 in-vitro [94]. Similarly, an adapter CAR (AdCAR) system whereby the CAR recognizes biotin and uses biotinylated antibodies as targeting modules has been used in NK-92 cells to target multiple tumour antigen in in-vitro models of non-Hodgkin’s lymphoma (NHL), mantle-cell lymphoma (MCL) and chronic lymphocytic leukaemia (CLL) [96].

Use of the NK-92 cell line is particularly beneficial as it removes the need for isolation of primary NK cells from healthy donors and has unlimited potential for cell number. However, as NK-92 cells are derived from non-Hodgkin’s lymphoma, the cells are normally irradiated prior to infusion into patients to prevent any possibility malignant expansion [87]. Therefore, a limitation of using these cells as a carrier of a CAR is that the irradiation prevents proliferation of the NK-92 cells and impacts their longevity [97,98].

## 5. Invariant NK T Cells

Taking this principle further, NK T cells are a type cytotoxic immune cell that share features of both NK cells and T cells. Both NK cells and NK T cells can recognise and destroy cancer cells without prior exposure to these cells. They differ in that NK cells contain cytoplasmic granules containing granzyme B and perforin but NK T cells do not [99]. NK cells express both Fc receptors and inhibitory receptors that are not present on NK T cells and similarly the NK T cells express a TCR, which is not present on NK cells.

A subset of NK T cells called ‘invariant NKT cells’ (iNKT cells) are able to recognise lipid antigens presented by CD1d on B cells, APCs and epithelial cells [100]. These lipid antigens are recognised through a very specific TCR expressed by the iNKT cells. Allogeneic iNKT cells have been shown to be protective against GvHD due to the production of IL-4 and promotion of a Th2-biased immune response, making them an attractive candidate for allogeneic CAR T cell therapy [101,102,103]. iNKT cells expressing a CD19 targeting CAR have shown both in-vitro and in-vivo efficacy against lymphoma cells expressing both CD19 and CD1d. In a murine model of CD1d+CD19+ B cell malignancy, the CAR19-iNKT cell-treated group displayed a significantly improved overall and tumour-free survival compared to those treated with conventional CAR19-T cells [104]. Anti-GD2 CAR-NKT cells have shown anti-tumour effects against neuroblastoma cells in-vitro and in-vivo. The NKT cells showed a lower toxicity profile compared to T cells expressing the same CAR, with the CAR T cells causing lethal GvHD in humanised NSG mice [105].

A phase 1 trial is currently underway assessing the safety of a GD2 CAR and IL-15 expressing autologous NKT Cells to treat children with neuroblastoma (NCT03294954). Similarly, a CD19 NKT CAR produced from allogeneic cells is currently being assessed in a phase 1 trial for the treatment of B cell malignancies (NCT03774654) (Table 1).

## 6. Genome Editing of αβ T Cells

Conventional αβT cells can also be used as the basis to generate allogeneic CAR T cells. However, they must first be modified to inactivate their capacity to induce TCR-dependent GvHD. To ensure this, the αβ TCR must be removed from the cell surface to prevent the T cells reacting against the host. To achieve this, it is sufficient to knockout the gene encoding the alpha chain (TRAC). This prevents the association of αβ heterodimer with CD3γ,δ,ε and ζ subunits, a process that is essential for cell surface expression of this complex.

Several genome editing techniques have been used to disrupt the αβ TCR including Zinc finger nucleases (ZFNs), CRISPR/Cas9 and Transcription activator-like effector nuclease (TALEN). However, these techniques do not result in complete knock out of both alleles in all target cells, with the result that some cells expressing residual αβ TCR may be introduced into the patient. This could result in GvHD, so additional rounds of selection are often required [106]. There is also a risk of off target genome cleavage and/or translocation formation, which could theoretically confer oncogenic potential on the CAR T cells [107]. The production of αβ CAR T cells is usually performed using a two-step technique where the gene encoding the CAR is integrated into the T cell using a viral vector followed by nuclease disruption of the TRAC loci (Figure 2).

This degree of genetic modification of primary cells can be problematic due to manipulation-related toxicity and poor transduction efficiency. However, a technique has been developed recently whereby the CAR construct is integrated into the TRAC locus. This simultaneously disrupts the αβ TCR and promotes CAR expression, meaning there are no CAR T cells produced that express the TCR [108,109]. This same result has also been achieved by combining the CAR construct and the CRISPR/Cas9 guide RNA in the same vector. This results in knock out of the αβ TCR in cells expressing the CAR (Figure 2) [110]. There is evidence that these αβ TCR negative cells have superior anti-tumour activity compared to traditional CAR T cells containing the native αβ TCR. Sadelain argued that this may be due to a reduction in tonic signalling from the TCR [107]. However, long term in-vivo persistence of TCR-KO cells appears to be lower than those expressing the native TCR [111]. TALEN-mediated disruption of the TRAC locus has also been combined with PD-1 disruption to improve the activity of the CAR T cells [112]. Similarly, CRISPR/Cas9 editing of the TRAC locus has been combined with CD7 disruption to produce a fratricide resistant “off-the shelf” CAR T cell [113]. Another approach to prevent GvHD is to express an inhibitory form of the TCR. The truncated CD3ζ called TIM (TCR inhibitory molecule) prevents TCR mediated cytotoxic activity against the host [114] (Figure 2). This strategy is currently being tested in a phase 1 study involving allogeneic CAR T cells targeted against NKG2D ligands (NCT03692429) (Table 1). As the elimination of the TCRαβ+ population is paramount for the safety of an off-the-shelf CAR T cell, new methods are being developed to efficiently deplete this population. Juillerat et al. have developed a new method to produce an ultra-pure TCRαβ^–^ population by transiently expressing a CD3 targeting CAR on the T cells using mRNA electroporation [115,116]. This temporarily redirects the CAR T cells to kill any CAR T cells expressing the αβTCR complex. Although this method has only been used in a proof of principle study, it has the potential to be used in clinical CAR T cell manufacture [115].

Although these approaches can prevent GvHD, they do not reduce immune rejection by the host. To overcome this the HLA-class 1 molecules can be removed from the T cells [117]. To remove HLA-class 1 the Beta-2 microglobulin (B2M) gene locus can be disrupted as it is required for HLA-class 1 expression (Figure 2) [118,119,120]. Additionally, mechanisms utilised by viruses to evade immune recognition have been investigated. One such approach used gamma-retroviral transduction of human T cells to introduce E3 ligase transgenes into the cells. These ligase are used by human herpes virus-8 (HHV8) to selectively ubiquinate MHC proteins to cause their trafficking away from the cell surface [121,122]. Introduction of these ligase reduces NK cell targeting of CAR T cells in vitro by removing the MHC proteins from the surface of the CAR T cells. This prevents the NK cells recognising the MHC mismatch, thus reducing NK cell mediated destruction of the CAR T cells [123].

It is however possible that removal of the HLA will result in NK cell recognition and destruction of these modified T cells [118]. CD19 targeting CAR T cells with both a TRAC and B2M gene disrupted by CRISPR/Cas9 are currently under clinical investigation for the treatment of relapsed or refractory CD19+ leukaemia and lymphoma (NCT03166878). Activated T cells also express HLA class II, which can be recognised by the host immune system and cause CAR T cell destruction [124]. Triple knockout of HLA class I, HLA class II, and T cell receptor by triple CRISPR/Cas9 targeting of the *B2M, CIITA*, and *TRAC* genes result in better persistence in in-vivo models compared to double knock out B2M and TRAC [119].

Sommer et al. [125] developed allogeneic CAR T cells from healthy donors that expressed the Flt3 CAR to target AML. These cells were modified using TALENs to remove TRAC and CD52 (Figure 2) [125]. CD52 is a glycoprotein upregulated on the cell surface of multiple types of immune cells [126]. Anti-CD52 antibodies can be used to deplete host immune cells to allow engraftment of the CAR T cells. Removal of CD52 from the CAR T cells makes them resistant to this lymphodepletion so that they can persist whilst the host immune cells are destroyed [125]. These KO cells display equivalent effector function compared to the non-gene edited equivalents and caused a reduction in tumour burden in an orthotopic mouse model of AML compared to control T cells [125]. This group has also developed an allogeneic αβ CAR T cells targeting BCMA in multiple myeloma. These cells are currently being investigated in clinical trials (NCT04093596) [127]. Similarly, a CD123 (a glycoprotein expressed on B cells) targeting CAR with TALEN-mediated inactivation of TRAC is also being investigated in a phase 1 study in AML (NCT03114670) [128] (Table 1).

## 7. Induced Pluripotent Stem Cells

Another potential source of CAR T cells are induced pluripotent stem cells (iPSCs). IPSCs can be produced from a starting cell type that is reprogrammed into a pluripotent cell using a cocktail of small molecules and transcription factors [129,130,131]. Any somatic cell can be used as the basis for an iPSC, however it has been shown that using a mature lymphoid cell as the starting cell results in a greater number of CD4^+^CD8^+^ T cells after iPSC differentiation [132,133].

Single clones from these reprogrammed cells are selected and expanded to produce a master iPSC line. These cells can then be differentiated into T cells, which can be further modified to express the CAR. One of the most attractive features of iPSCs is their unlimited potential for division, overcoming the problem of expanding primary T cells. iPSCs are also more amenable to genetic modification and these changes can be made to parental cells to give rise to a modified cell line [130].

Like αβ T cells, T cells derived from iPSCs must be modified to remove molecules that can mediate GvHD and rejection of the T cells. Due to the nature of these cells, repeated genetic modification technically is not a major problem, though ensuring that the gene inserts do not activate a silent proto-oncogene is a major consideration. Once modified a single cell can be used to generate a clonal population with the desired genetic knockouts. In addition, iPSCs derived from T cells already contain the rearranged TCR locus of the parental cell, thus iPSCs with a particular TCR specificity can be selected for [134].

The first clinical trial testing this approach is now underway using iPSC derived CD19 CAR T cells in patients with B cell malignancies (NCT04629729) [135]. This approach can also be used to generate CAR NK cells in addition to CAR T cells [136].

## 8. Conclusions

Unfortunately, the successes of CAR T cells is yet to be widely clinically applicable. This is due to many factors including lack of appropriate tumour specific antigens, the immunosuppressive nature of the tumour microenvironment, variability in product quality and high costs (previously discussed by Cutmore et al. [137]. Use of allogeneic CAR expressing cells could overcome some of these obstacles. Allogeneic CAR products can be made on a more industrial scale, making the production pipeline cheaper, thus rendering the treatment more accessible to healthcare systems and more patients. Use of allogeneic cells from healthy donors also means that the cellular product is of a higher quality and would be available for larger numbers of patients. It would also allow for re-dosing and more standardised dosing and activity.

This is particularly true when using cell lines such as NK-92 as they represent an unlimited supply of donor cells. Use of allogeneic cells also means that the CAR T or CAR NK cells can be produced in bulk and stored. This means that they are available immediately and patients do not have to wait two–three weeks to receive the treatment. This is especially important in cancers that progress rapidly, as using current manufacturing protocols the disease has often progressed too much by the time the CAR T cells are ready to be infused. Importantly, the use of allogeneic CAR T cells would also allow for targeting of multiple antigens sequentially or simultaneously as one batch of cells could be transduced with multiple CAR constructs as there is no limit of cell numbers obtained.

The future of CAR T cell therapy looks promising. Advances in technology and gene editing techniques could make allogeneic “off the shelf” CAR T cells available in the near future and could revolutionise the treatment of cancers.

## Figures and Tables

**Figure 1 cancers-13-01926-f001:**
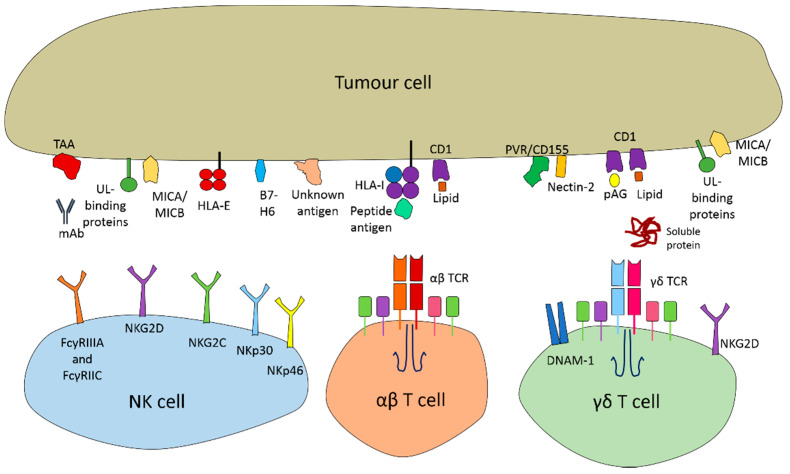
Diagram illustrating some of the main receptors present on NK cells, αβ T cells and γδ T cells and their corresponding ligands that are involved in tumour cell recognition and killing. NK cells express a variety of receptors that allow them to recognize tumour cells, the FcyR receptors recognised the constant region of antibodies what bind to TAA’s on tumour cells. NKG2D expressed on both NK cells and γδ T cells, has a variety of ligands, including UL16-binding proteins and MICA/B. NKG2C recognizes HLA-E expressed on tumour cells. NKp30 can bind B7-H6 and NKp46 mediates tumour directed killing via an unknown ligand. αβ T cells express a αβ TCR which recognises peptide antigens that are sampled from inside the tumour and presented on HLA-I molecules. Both αβ and γδ TCRs can also recognise lipid antigens expressed on CD1. Additionally, γδ T cells also express DNAM-1 which recognises PVR/CD155 and Nectin-2. Unlike the αβ TCR, the γδ TCR can recognise soluble protein and is not restricted to recognizing antigen presented on HLA.

**Figure 2 cancers-13-01926-f002:**
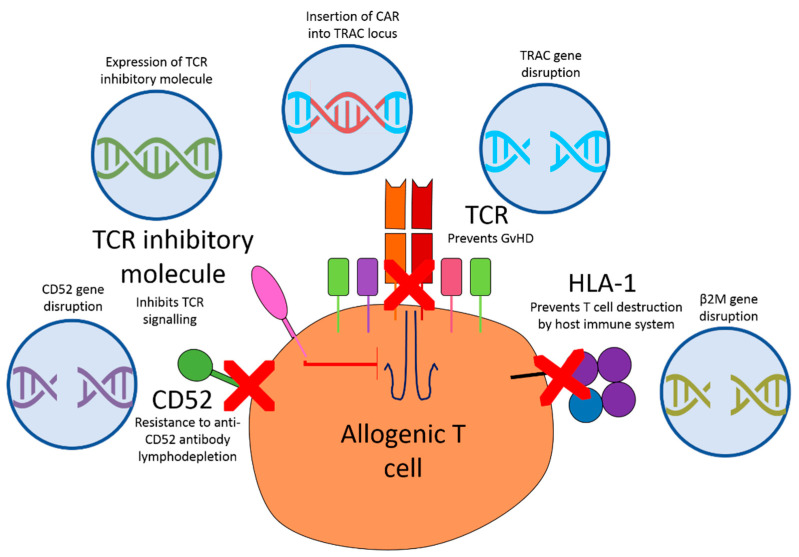
Diagram illustrating genetic modifications that can be made to donor αβ T cells to make allogenic CAR T cell therapy possible. To prevent GvHD the αβ TCR must be disrupted. This can be achieved by inserting the CAR construct into the TRAC locus or by knocking down the TRAC gene using TALENs, ZFNs or CRISPR/Cas9 to prevent TCR expression. Additionally, a TCR inhibitory molecule can introduced to the cells to prevent TCR signalling. CD52 gene disruption has also been used to provide the allogenic T cells with resistance to lymphodepletion. To prevent destruction of the allogeneic cells by the host immune system, the B2M locus can be disrupted. This prevents the formation of HLA-1 molecules on the surface of the T cells and prevents them from being recognised as foreign.

**Table 1 cancers-13-01926-t001:** Past and current clinical trials involving allogeneic CAR cell therapies for the treatment of cancer.

Cell Type	CAR Target	Modification	Disease	Trial Number	Gene Editing Technology
γδ T cells	CD19		Leukaemia, Lymphoma	NCT02656147	NS
γδ T cells	NKG2D ligands		Colorectal Cancer, Triple Negative Breast Cancer, Sarcoma, Nasopharyngeal Carcinoma, Prostate Cancer, Gastric Cancer	NCT04107142	NS
T cells	CD7	TRAC disrupted	T cell Leukaemia, T cell Lymphoma	NCT04264078	NS
T cells	CD19	TRAC and CD52 genes have been disrupted	B-cell Acute Lymphoblastic Leukaemia (B-ALL)	NCT04166838	NS
T cells	CD19	Unknown	B-cell Acute Lymphoblastic Leukaemia, B-cell Lymphoma	NCT04264039	NS
T cells	CD7	Unknown	T Acute Lymphoblastic Leukaemia and Lymphoma	NCT04620655	NS
T cells	CD7	Unknown	Haematologic Malignancies	NCT04538599	NS
T cells	CD123	Unknown	Blastic Plasmacytoid Dendritic Cell Neoplasm (BPDCN)	NCT03203369	TALENs
T cells	BCMA	Unknown	Relapsed/Refractory Multiple Myeloma	NCT04601935	NS
T cells	CD19	Epstein-Barr Virus Specific Cytotoxic T-Lymphocytes (EBV-CTLs)	Acute Lymphocytic Leukaemia, Lymphoma	NCT01430390	NS
T cells	CD19	TRAC and CD52 genes have been disrupted	Advanced Lymphoid Malignancies	NCT02735083	NS
T cells	CD19	Unknown	Leukaemia	NCT02799550	NS
T cells	CD19	TRAC disrupted	Lymphoma	NCT04637763	CRISPR-Cas9
T cells	CD19	TRAC and CD52 genes have been disrupted	Relapsed/Refractory Large B Cell Lymphoma	NCT04416984	CRISPR-Cas9
T cells	CD19	TRAC disrupted	Non-Hodgkin Lymphoma, B-cell Acute Lymphoblastic Leukaemia	NCT03666000	NS
T cells	BCMA	Unknown	Multiple Myeloma	NCT03752541	CRISPR-Cas9
T cells	CD123	TRAC and CD52 genes have been disrupted	Acute Myeloid Leukaemia	NCT04106076	TALENs
T cells	CD123	TRAC and CD52 genes have been disrupted	Relapsed/Refractory Acute Myeloid Leukaemia	NCT03190278	TALENs
T cells	CD123	TRAC and CD52 genes have been disrupted	Blastic Plasmacytoid Dendritic Cell Neoplasm (BPDCN)	NCT03203369	TALENs
T cells	CD19	TRAC and CD52 genes have been disrupted	Adult Patients With Relapsed/Refractory B-cell Acute Lymphoblastic Leukaemia	NCT02746952	TALENs
T cells	CD19	TRAC and CD52 genes have been disrupted	Advanced Lymphoid Malignancies	NCT02735083	TALENs
T cells	CD19	TRAC and CD52 genes have been disrupted	Paediatric Relapsed/Refractory B-cell Acute Lymphoblastic Leukaemia	NCT02808442	TALENs
T cells	CD19	TRAC and CD52 genes have been disrupted	Acute Lymphoblastic Leukaemia (ALL), Non-Hodgkin Lymphoma (NHL)	NCT03229876	CRISPR-Cas9
T cells	CD19	TRAC and CD52 genes have been disrupted	B Cell Leukaemia, B Cell Lymphoma	NCT03166878	CRISPR-Cas9
T cells	CD19	TRAC and CD52 genes have been disrupted	Relapsed/Refractory Lymphoma	NCT03939026	TALENs
T cells	BCMA	TRAC and CD52 genes have been disrupted	Relapsed/Refractory Multiple Myeloma	NCT04093596	TALENs
T cells	CD19	MHC-1 Knockout by B2M disruption and CAR inserted into TRAC locus	B-cell Malignancy, Non-Hodgkin Lymphoma, B-cell Lymphoma	NCT04035434	CRISPR-Cas9
T cells	BCMA	MHC-1 Knockout by B2M disruption and CAR inserted into TRAC locus	Multiple Myeloma	NCT04244656	CRISPR-Cas9
T cells	CD19	Unknown	Acute Lymphoblastic Leukaemia	NCT04154709	CRISPR-Cas9
T cells	CD22	TRAC disrupted	B-cell Acute Lymphoblastic Leukaemia	NCT04150497	TALENs
T cells	CD19	Unknown	Diffuse Large B-cell Lymphoma	NCT04026100	CRISPR-Cas9
T cells	CD19 + CD20 or CD19+CD22	Unknown	B Cell Leukaemia, B Cell Lymphoma	NCT03398967	CRISPR-Cas9
T cells	CS1	TRAC disrupted	Relapsed/Refractory Multiple Myeloma	NCT04142619	TALENs
T cells	CD19	Unknown	B-cell Acute Lymphoblastic Leukaemia, B-cell Lymphoma	NCT04264039	NS
T cells	CD30	Epstein-Barr Virus-Specific T Lymphocytes	Lymphoma	NCT04288726	NS
T cells	CD22	Unknown	B Cell Malignancy	NCT04601181	NS
T cells	CD19	Unknown	B Cell Malignancy	NCT04384393	NS
T cells	Unknown	Unknown	Ovarian Cancer	NCT00019136	NS
T cells	CD70	B2M KO and CAR insertion into TRAC locus	T Cell Lymphoma	NCT04502446	CRISPR-Cas9
T cells	CD19	Unknown	ALL, Childhood B-Cell	NCT04173988	NS
T cells	GD2	Tri-virus specific cytotoxic T cells	Neuroblastoma	NCT01460901	NS
T cells	CD20	CAR inserted into TCR locus	Non-Hodgkin’s Lymphoma,	NCT04030195	NS
T cells	CD19	TRAC locus disrupted	B Acute Lymphoblastic Leukaemia	NCT04557436	CRISPR-Cas9
T cells	Cd19	Unknown	B-cell Acute Lymphoblastic Leukaemia, B-cell Non Hodgkin Lymphoma	NCT04544592	NS
T cells	CD20	Unknown	Diffuse Large B Cell Lymphoma, Follicular Lymphoma, Mantle Cell Lymphoma, Small Lymphocytic Lymphoma	NCT04176913	NS
T cells	CD19	Unknown	B cell Lymphoma	NCT02992834	NS
T cells	CD19	Donor-derived EBV-specific cytotoxic T cells	Acute Lymphoblastic Leukaemia	NCT01195480	NS
T cells	CD70	MHC-1 Knockout by B2M disruption and CAR inserted into TRAC locus	Renal Cell Carcinoma	NCT04438083	CRISPR-Cas9
T cells	NKG2D ligands		Colorectal Cancer	NCT03692429	Non-gene edited peptide-based technology
NKT cells	CD19		Lymphoma	NCT03774654	NS
NK cells	CD19	NK-92 cells	Leukemia	NCT02892695	NS
NK cells	CD7	pNK cells	Leukemia	NCT02742727	NS
NK cells	CD19	CB-NK	CD19 Positive Lymphoma	NCT03579927	NS
NK cells	ErbB2/Her2	NK-92 cells	Glioblastoma	NCT03383978	NS
NK cells	Unknown	NK-92 with Chimeric Costimulatory Converting Receptor (CCCR)	Non-small Cell Lung Cancer	NCT03656705	NS
NK cells	CD33	NK-92	Leukaemia	NCT02944162	NS
NK cells	BCMA	NK-92	Multiple Myeloma	NCT03940833	NS
NK cells	CD19	NK-92	Leukaemia, Lymphoma	NCT02892695	NS
NK cells	NKG2D ligands		Solid Tumours	NCT03415100	NS
NK cells	NKG2D ligands	Membrane IL-15	Relapsed/Refractory AML	NCT04623944	NS

Abbreviations: TRAC, T cell receptor alpha locus. NS, Not stated.

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
