# Peer review of "Current Perspectives on the Use of off the Shelf CAR-T/NK Cells for the Treatment of Cancer"

_cancers, 2021, doi:10.3390/cancers13081926_

Round 1
Reviewer 1 Report
The review focused on different cell sources for the production of allogeneic CAR T cells.
There were some minor typos.
Line 62 'in the presence of IL-2 then re-infused'
Line 66- simplify to make it easier to read.
line 142- 'for effectiveness of CAR T cell immunotherapy;?
line 318- 'several'
line 359- reference 110 refers to a Flt 3 targeted CAR. But the section follows on from describing an anti-BCMA CAR, so it is confusing as to why it is tested in an AML model. Suggest introduce the Flt 3 CAR described in the paper.
Are there any theoretical safety concerns to using NK-92 cell line? Is there any data on persistence?
Author Response
We thank reviewer 1 for their helpful comments and critique. We address each in turn:
Line 62 'in the presence of IL-2 then re-infused'- corrected
Line 66- simplify to make it easier to read- we have changed it to "In later studies, tumour infiltrating lymphocytes (TIL) were isolated from patients and expanded ex-vivo before reinfusing them back into the same patient at higher numbers
line 142- 'for effectiveness of CAR T cell immunotherapy;- we cannot identify the typo
line 318- 'several'-corrected
line 359- reference 110 refers to a Flt 3 targeted CAR. But the section follows on from describing an anti-BCMA CAR, so it is confusing as to why it is tested in an AML model. Suggest introduce the Flt 3 CAR described in the paper. -thank you for identifying our error in our text now beginning at line 442. We had referred to one of this group's papers while referencing another. We have now corrected this by inserting the following amended text:
"Sommer et al (126) developed allogeneic CAR T cells from healthy donors that expressed the Flt3 CAR to target AML. These cells were modified using TALENs to remove TRAC and CD52 (Figure 2) (126). CD52 is a glycoprotein upregulated on the cell surface of multiple types of immune cells (127).. CD52 is removed to improve CAR T cell persistence in patients following anti-CD52 antibody-based lymphodepletion of alloreactive immune cells (126)). These KO cells display equivalent effector function compared to the non-gene edited equivalents and caused a reduction in tumour burden in an orthotopic mouse model of AML compared to control T cells (126). This group has also developed an allogeneic αβ CAR T cells targeting BCMA in multiple myeloma. These cells are currently being investigated in clinical trials (NCT04093596) (128)."
Are there any theoretical safety concerns to using NK-92 cell line? We have added "However, as NK-92 cells are derived from non-Hogkins lymphoma, the cells are normally irradiated prior to infusion into patients to prevent any possibility of malignant expansion (97). Therefore, a limitation of using these cells as a carrier of a CAR is that the irradiation prevents proliferation of the NK-92 cells and impacts their longevity (98, 99) "
Is there any data on persistence?-we are unaware of in vivo data
Reviewer 2 Report
The review article by LC Cutmore and JF Marshall offers a comprehensive state of the art of the CAR-T cell biology, highlighting the wide range of opportunities available for maximizing clinical outcome reducing risks and side effects of CAR cell therapy for solid and liquid tumors. In particular, their work summarizes and discuss some interesting and burning topics related to the choice of immune effector cells and the possibility of manipulating them to make this therapy more “Universal”.
Nothing is really new, but is a well written manuscript and quite up-to date. Quite …, because many of the article cited were published some time ago and perhaps current advances in the field may not be adequately reflected. Their literature search related to CAR T is informative, well-presented and homogenously distributed in the text, but there are only few papers published in 2018-2020, both as reviews cited as well as original papers.
Few missing papers from the reference list that are important and add will value to this review. So please go through latest literature and cite their information too.
Furthermore, more effort should be dedicated to the legends of figures 1 and 2. A detailed explanation is required to guide the reader for a better comprehension of concepts illustrated in the figures.
Overall, in my opinion, this manuscript can be accepted after minor revision as I suggested.
Author Response
We thank Reviewer 2 for their helpful comments which we deal with systematically below.
Quite …, because many of the article cited were published some time ago and perhaps current advances in the field may not be adequately reflected. -we have added over 15 additional more recent reports from 2020 and 2021
Furthermore, more effort should be dedicated to the legends of figures 1 and 2. A detailed explanation is required to guide the reader for a better comprehension of concepts illustrated in the figures.- we have provided more informative legends for both figures
Reviewer 3 Report
The review article reports the state of the art of CAR-T cell biology and functioning, and possible " universal" strategies to increase efficacy and tolerability of this kind of immunotherapy. Likely, the authors should integrate the paper with some recent publication in the fields that might augment the relevance of this review. Legends to figures are not so informative for a review of general interest among oncologists.
Author Response
We thank reviewer 3 for their helpful comments that we address systematically below.
Likely, the authors should integrate the paper with some recent publication in the fields that might augment the relevance of this review.-we have updated the references with more recent publications from the last 2 years.
Legends to figures are not so informative for a review of general interest among oncologists.- we have provided longer and more detiled legends to both figures